# Parking Lot Occupancy Detection with Improved MobileNetV3

**DOI:** 10.3390/s23177642

**Published:** 2023-09-03

**Authors:** Yusufbek Yuldashev, Mukhriddin Mukhiddinov, Akmalbek Bobomirzaevich Abdusalomov, Rashid Nasimov, Jinsoo Cho

**Affiliations:** 1Department of Computer Engineering, Gachon University, Seongnam-si 13120, Republic of Korea; yusufbek02106@gmail.com (Y.Y.); mukhiddinov18@gachon.ac.kr (M.M.); bobomirzaevich@gmail.com (A.B.A.); 2Department of Artificial Intelligence, Tashkent State University of Economics, Tashkent 100066, Uzbekistan; rashid.nasimov@tsue.uz

**Keywords:** deep learning, classification, convolutional neural networks, MobileNetV3, parking space management, parking lot

## Abstract

In recent years, parking lot management systems have garnered significant research attention, particularly concerning the application of deep learning techniques. Numerous approaches have emerged for tackling parking lot occupancy challenges using deep learning models. This study contributes to the field by addressing a critical aspect of parking lot management systems: accurate vehicle occupancy determination in specific parking spaces. We propose an advanced solution by harnessing an optimized MobileNetV3 model with custom architectural enhancements, trained on the CNRPark-EXT and PKLOT datasets. The model processes individual parking space patches from real-time video feeds, providing occupancy classification for each patch, identifying occupied or available spaces. Our architectural modifications include the integration of a convolutional block attention mechanism in place of the native attention module and the adoption of blueprint separable convolutions instead of the traditional depth-wise separable convolutions. In terms of performance, our proposed model exhibits superior results when benchmarked against state-of-the-art methods. Achieving an exceptional area under the ROC curve (AUC) value of 0.99 for most experiments with the PKLot dataset, our enhanced MobileNetV3 showcases its exceptional discriminatory power in binary classification. Benchmarked against the CarNet and mAlexNet models, representative of previous state-of-the-art solutions, our proposed model showcases exceptional performance. During evaluations using the combined CNRPark-EXT and PKLot datasets, the proposed model attains an impressive average accuracy of 98.01%, while CarNet achieves 97.03%. Beyond achieving high accuracy and precision comparable to previous models, the proposed model exhibits promise for real-time applications. This work contributes to the advancement of parking lot occupancy detection by offering a robust and efficient solution with implications for urban mobility enhancement and resource optimization.

## 1. Introduction

The problem of parking has become increasingly problematic as the number of cars on the roads has increased, particularly in urban areas. Therefore, there is a strong demand for effective parking lot management systems that can address these problems in real time. With limited parking availability and the ever-growing number of vehicles, traditional parking management approaches are proving inadequate in ensuring optimal space utilization and reducing congestion. Deep learning techniques, particularly convolutional neural networks (CNNs), have gained attention for their potential to transform parking management. These methods offer the promise of accurate occupancy detection, which is fundamental for making informed decisions regarding space allocation, traffic flow optimization, and overall urban planning.

Several studies have proposed deep learning techniques for parking lot management, with a specific focus on three types of problems: automatic parking space position detection, individual parking space classification, and vehicle detection and counting [1]. The motivation behind developing a model for parking lot occupancy detection is to address the need for efficient management of parking spaces. By accurately determining the occupancy status of parking lots, it becomes possible to optimize parking resource utilization, enhance traffic management, and improve the overall parking experience for users.

However, the successful integration of deep learning in parking management necessitates a profound understanding of the unique challenges posed by this domain. Parking scenarios introduce complexities such as varying lighting conditions, diverse vehicle types, occlusions, and the requirement for real-time response. These challenges demand tailored solutions that can reliably function across a spectrum of conditions, providing accurate occupancy detection while accommodating the dynamic nature of parking environments. Existing methods for parking lot occupancy detection often rely on conventional computer vision techniques or shallow machine learning models, which struggle to achieve high accuracy in complex parking scenarios. These methods lack the ability to handle some of the aforementioned problems.

In this paper, we address these challenges by proposing an enhanced MobileNetV3 architecture customized for the nuanced demands of parking lot occupancy detection. By leveraging the architectural efficiency of MobileNetV3 [2] and introducing domain-specific modifications, we aim to mitigate the complexities inherent to parking management scenarios. Although the MobileNetV3 architecture has demonstrated significant efficiency gains and high accuracy in various computer vision tasks, its application to parking lot occupancy detection poses unique challenges. In the context of parking lot occupancy detection, the original MobileNetV3 encounters limitations related to handling varying lighting conditions, dealing with occlusions, and distinguishing between different vehicle types. These challenges stem from the specific characteristics of parking lot images, including complex backgrounds, varying perspectives, and the need to accurately identify small, partially occluded objects. Our research addresses these limitations by introducing key modifications to the MobileNetV3 architecture tailored for parking lot occupancy detection. This modified version incorporates several architectural improvements, including the use of a Leaky-ReLU6 [3] activation function for the shallow part of the MobileNetV3 model, the replacement of the squeeze-and-excitation module [4] with the convolution block attention module [5], and the replacement of the depth-wise separable convolutions with blueprint separable convolutions [6]. We treat the automatic detection of vacant spaces as a binary classification problem and train and test the improved model on widely used parking management datasets such as CNRPark-EXT [7] and PKLOT [8]. The proposed model processes individual parking spaces and classifies them as vacant or occupied. The incoming real-time video feed frame is processed to obtain individual parking spaces. The proposed model exhibits superior performance compared to previous state-of-the-art models in terms of accuracy and precision and demonstrates its capability to function in real time.

The industrial significance of our approach lies in its practical applications within the rapidly growing field of smart cities and intelligent transportation systems. Our modified MobileNetV3 model addresses key challenges in parking management, contributing to reduced congestion, improved user experiences, and optimized parking resource utilization. With real-time and accurate parking occupancy detection, cities can implement responsive parking guidance systems, enabling drivers to quickly locate available parking spots.

The main contributions of this study are as follows:Novel model outperforming state-of-the-art models: We propose and develop a novel model that achieves a substantial advancement over existing state-of-the-art models in terms of both accuracy and AUC score. Importantly, this superior performance is achieved while ensuring real-time functionality, making our model highly suitable for practical applications.Enhancements to MobileNetV3 architecture: We enhance the performance of the MobileNetV3 architecture through a series of strategic modifications. Firstly, we introduce a novel activation function that contributes to improved accuracy and precision. Additionally, we replace the traditional squeeze-and-excitation (SE) module with a Convolution Block Attention Module (CBAM), a change that refines the model’s ability to focus on salient features. Moreover, we optimize the depth-wise convolution block by adopting blueprint separable convolutions, resulting in a model architecture that is more efficient and effective for parking management tasks.Improved generalization and small object detection: Our enhanced MobileNetV3 model exhibits notable improvements in its architecture. These modifications empower the model to better identify essential aspects of images, pay attention to small objects within the image, and achieve increased generalization capability. These enhancements collectively contribute to superior performance in parking lot occupancy detection tasks.Practical significance: The contributions outlined above hold significant implications for real-world parking management scenarios. Our model’s elevated accuracy, coupled with its capacity for real-time operation, has the potential to revolutionize parking lot occupancy detection. By honing in on crucial image components and effectively detecting small objects, our model proves to be a valuable asset for optimizing parking resource utilization, alleviating traffic congestion, and ultimately enhancing the efficiency of parking management systems.

The remainder of this paper is organized as follows. Section 2 reviews the literature concerning the MobileNet models’ family and parking space classifications. Section 3 describes the datasets used in the experiments. Section 4 and Section 5 discuss the proposed parking management approach and present the experimental results and analyses, respectively. Section 6 provides an overview of the research findings and suggests potential areas for future investigation.

## 2. Background and Related Work

In this section, we briefly discuss the models in the MobileNet family in Section 2.1 and discuss feature extraction and deep learning-based solutions for the problem of parking space classification in Section 2.2.

### 2.1. MobileNet Model Family

Recently, deep learning has significantly contributed to the field of computer vision. This has led to the development of diverse technologies, including object detection, classification, and segmentation models, which can be accessed through cloud-based services on internet-connected devices. However, the implementation of these models and technologies on mobile and embedded devices poses several challenges: the models must operate with high accuracy and speed while being mindful of the limited computational power and resources available. Therefore, models that satisfy this criterion are critical. An efficient model family called MobileNet was introduced in 2017 [9] and was specifically designed for use on mobile and embedded devices. These models exhibited better performance in terms of latency, size, and speed than other cutting-edge models. Although there is a slight latency in the output performance, this tradeoff is acceptable when the model can be deployed on a mobile or edge device for real-time offline detection. MobileNet is a family of low-latency, low-power, and small computer vision model architectures designed to maximize accuracy while satisfying the resource constraints of on-device or embedded applications. Classification, detection, and segmentation tasks can be performed using these models, as with other large-scale models. 

MobileNetV1 [9], the first model in the MobileNet family, incorporated a new feature called depth-wise separable convolution, which can drastically reduce the number of required parameters compared with other architectures that use regular convolutions. The architecture of the MobileNetV1 model is presented in Figure 1, containing several depth-wise separable convolutions that decrease the number of operations required by the model for forward and backward propagation.

The shape of the input image is 224 × 224 × 3 and 3 × 3 × 3 convolution is applied onto it firstly. The resulting feature map is 32 × 112 × 112 and in the first convolution layer, C1, 3 × 3 × 1 convolution filter is applied. Two operations, depth-wise and point-wise convolution, constitute the depth-wise separable convolution. Depth-wise separable convolution is based on the concept of separating a filter’s depth and spatial dimensions; this is the reason why it is called “separable.” This concept involves isolating the depth dimension from the horizontal dimension, resulting in depth-wise separable convolution. This approach entails first performing depth-wise convolution and then applying a 1 × 1 filter across the depth dimension. The reduction in the number of parameters achieved with this convolution is impressive. For instance, creating a single channel necessitates 3 × 3 × 3 parameters for depth-wise convolution and only 1 × 3 parameters for additional convolution in the depth dimension. However, for the same number of output channels in the regular convolution, three 3 × 3 × 3 filters are required, resulting in a total requirement of 81 parameters. In contrast, for the depth-wise separable convolution, only three 1 × 3 depth filters are required, resulting in a total of 36 parameters. Figure 2 illustrates depth-wise separable convolution, involving point-wise convolution following depth-wise convolution.

The process of depth-wise convolution involves applying one convolution per input channel individually, resulting in the same number of output channels as input channels; its computational cost is calculated as Df2 × M × Dk2. Point-wise convolution, on the other hand, is a type of convolution with a kernel size of 1 × 1 that merges the features created by depth-wise convolution, whose computational cost is computed as M × N × Df2.

MobileNetV2 [10] uses depth-wise separable convolutions in its design, with its main block comprising three convolutional layers, as shown in Figure 3.

The last two convolutions are the same as those in MobileNetV1, consisting of one depth-wise and one 1 × 1 point-wise convolution. In contrast to MobileNetV1, the point-wise convolution in MobileNetV2 reduces the number of channels, thereby earning it the name “projection layer,” as high-dimensional data are projected onto a tensor with fewer dimensions. This convolutional layer is also referred to as the bottleneck layer because it minimizes the amount of data flowing through the network. MobileNetV2 also introduces a new 1 × 1 convolution in its design, which increases the number of channels in the data before they are subjected to depth-wise convolution. The expansion factor, a hyperparameter that determines the extent to which the number of channels is expanded, is set to six by default and requires experimental verification. The use of a residual connection, as in ResNet, is another new feature of MobileNetV2 that facilitates the flow of gradients within the network.

MobileNetV3 is the latest iteration in the MobileNet family of convolutional neural networks (CNNs), which incorporates squeeze-and-excitation (SE) blocks in the initial building blocks taken from MobileNetV2. Traditional convolutional layers in CNNs treat each channel equally; however, SE blocks compute the output by considering the relevance of each channel, as shown in Figure 4.

Each channel is first compressed into one numeric value by the SE block; this numeric value is then fed into a two-layer feed-forward network that calculates the weights for each channel. In MobileNetV3, SE blocks are applied in parallel to the residual layers to assign different weights to various channels in the input when creating the output feature maps, leading to improved accuracy.

MobileNetV3 also introduces two improvements over MobileNetV2: layer removal and the use of swish non-linearity.

(a)Layer removal: The 1 × 1 expansion layer, obtained from the inverted residual unit and transported along the pooling layer in the last block of MobileNetV2, uses 1 × 1 feature maps rather than 7 × 7 feature maps, making it efficient in terms of computation and latency. Consequently, the projection and filtering layers of the prior bottleneck layer may be eliminated, as shown in Figure 5.

(b)Use of swish non-linearity: Swish non-linearity is defined as

swish x = x*σ(x) (1)

Swish nonlinearity has been proven to enhance accuracy. Nevertheless, the creators of MobileNetV3 replaced the sigmoid function with the hard swish or h-swish because the sigmoid is computationally expensive, and computational expenditure must be minimized.
h-swish[x] = x*(ReLU6(x + 3))/6(2)

The research introducing MobileNetV3 defined two models: MobileNetV3-Large and MobileNetV3-Small, the structures of which are listed in Table 1.

### 2.2. Related Work on Parking Space Classification

Figure 6 illustrates two different states in which a parking spot can be: occupied or vacant. So, the parking space classification task can be designed as a binary classification task. The parking space classification task can be done via either traditional machine learning approaches or deep learning approaches. The initial image used as the input for the parking lot occupancy detection system is captured using a camera and consists of the entire parking lot. Before starting the classification process, individual parking spots are extracted from the whole parking lot image with provided parking spot locations.

With traditional machine learning approaches, we extract useful image features with different image preprocessing techniques, like histogram equalization or thresholding. After feature vector extraction, we train one of many classification algorithms, like a support vector machine or multilayer perceptron, with the feature vector and their ground truth labels. Various studies have proposed approaches that use feature extraction to classify individual parking spots.

Al-Kharusi et al. [11] introduced an intelligent parking management system exclusively reliant on conventional image processing methodologies. This system encompasses a series of operations including color space conversion, morphological operations (specifically dilation and erosion), thresholding techniques, edge detection algorithms, and a Hough transform. Ahrnbom et al. [12] devised a parking slot occupancy classifier by integrating Integral Channel Features with either Logistic Regression or Support Vector Machine. Initially, ten feature channels were extracted per input image, encompassing elements such as color channels in the LUV color space, gradient magnitude, and quantized gradient channels. Subsequently, feature vectors were efficiently computed from specific feature channels using the integral image approach. Finally, both logistic regression and support vector machine classifiers were trained and tested using the PKLot dataset. Furthermore, in de Almeida et al. [8], the authors not only provided a dataset but also tackled the issue by employing machine learning methodologies. They utilized their dataset, comprising around 700,000 images of parking spaces from multiple cameras in parking lots, to train Support Vector Machine (SVM) classifiers on diverse textural characteristics, including Local Binary Patterns (LBP), Local Phase Quantization (LPQ), and their derivatives. Additionally, they improved detection accuracy by employing combinations of SVMs and employing basic aggregation techniques, such as maximum or average, on the confidence scores generated by the classifiers.

Numerous scholars have acknowledged the constrained adaptability of handcrafted visual features, such as SIFT, SURF, and ORB, to effectively accommodate the intricacies of object appearance variations, which often exhibit high non-linearity, temporal fluctuations, and complexity. Intriguingly, pre-trained Convolutional Neural Networks (CNNs) have demonstrated remarkable efficacy as "off-the-shelf" feature extractors across a diverse array of visual recognition tasks, as evidenced by findings from Razavian et al. [13]. It should be noted that feature engineering is not applied in DL models because these models aim to discover how parking spots are represented, and the classifier is typically integrated into the DL model. Amato et al. [7] developed mAlexNet, a deep neural network tailored for parking occupancy classification. Through comprehensive evaluations on the PKLot and CNRPark-EXT datasets, mAlexNet demonstrates superior performance over AlexNet and LPQ by de Almeida et al. [8] in terms of both classification accuracy and area under the curve (AUC). Notably, despite being significantly smaller in size—approximately three orders of magnitude compared to the original AlexNet [14]—mAlexNet remains feasible for implementation on embedded platforms such as the Raspberry Pi 2 Model B. Nguyen et al. [15] proposed a modified version of mAlexNet along with their own dataset, HUSTPark, taken from two parking fields at the HUST campus. Their model was very compatible and small in size, but that came in the cost of reducing the accuracy of the model compared to original mAlexNet model.

In Nurullayev and Lee’s study [16], they introduced CarNet, a deep neural network (DNN) utilizing dilated convolutional neural networks to assess parking space occupancy status. CarNet takes as input a 54 × 32 RGB image representing a parking slot. Their experiments demonstrate CarNet’s superior performance compared to AlexNet [14] and other established deep learning architectures on the PKLot dataset. Furthermore, CarNet outperforms mAlexNet [7] on the CNRPark-EXT dataset. Despite achieving high precision and robustness, CarNet necessitates manual cropping of parking slot images from the overall parking lot input image. A detailed comparison between CarNet and our proposed approach is presented in the results section. 

In their paper, Xiao et al. [17] address the problem of whether a free parking spot is compatible with the mission of the ego vehicle by tackling parking spot classification based on the surround view camera system. The authors adapt the YOLOv4 object detection neural network, enhancing it with a novel polygon bounding box model suitable for various shaped parking spaces, including slanted parking slots. Notably, this study represents the first detailed investigation of parking spot detection and classification using fisheye cameras for auto valet parking scenarios. The proposed classification approach effectively distinguishes between regular, electric vehicle, and handicap parking spots. By considering both occupancy and suitability, this research contributes to more intelligent and context-aware parking guidance systems. Grbich et al. [18] introduced an algorithm for automatic parking slot detection as well as an occupancy classification model. Their approach contains several steps: detect cars in subsequent parking lot images with YOLO for approximately five minutes and extract bounding box centers; cluster and eliminate false detection bounding box centers with a clustering algorithm, and from cluster centers, detect bounding boxes of parking spaces; and perform the classification for detected parking spaces. Duong et al. [19] introduced an object detector for parking occupancy detection, OcpDet, based on RetinaNet, with its backbone, ResNet, replaced by MobileNet. They mainly emphasize scalability and reliability. Martynova et al. [20] created a new seasonal dataset for parking lot occupancy detection and developed a custom model based on EfficientNet-B0 for parking occupancy detection.

## 3. Datasets Used for Experiments

CNRPark-EXT and PKLot datasets were used in our experiments as the source of data. Table 2 shows the CNRPark-EXT and PKLot dataset features.

### 3.1. CNRPark-EXT Dataset

Amato et al. [7] developed the CNRPark-EXT dataset by extending the CNRPark dataset [21]. CNRPark-EXT is a comprehensive dataset designed for visual occupancy detection in parking lots. Figure 7 shows some examples from different camera perspectives and environmental conditions.

CNRPark-EXT contains approximately 150,000 labeled images (patches) representing both vacant and occupied parking spaces. The dataset is built on a parking lot with 164 parking spaces. It extends the original CNRPark dataset, which consisted of 12,000 images collected from two cameras during different days in July 2015. CNRPark-EXT is an additional subset, collected from November 2015 to February 2016, that significantly expands the dataset. It includes images captured by nine cameras with varying perspectives and angles of view. CNRPark-EXT captures diverse scenarios, including different light conditions, partial occlusions (due to obstacles like trees, lampposts, and other cars), and partial or global shadows on cars. The cameras in CNRPark-EXT cover a wide range of views, capturing parking spaces from different angles. The dataset provides a glimpse into the fields of view of the nine available cameras.

### 3.2. PKLot Dataset

The PKLot dataset is a robust collection designed specifically for parking lot classification developed by Almeida et al. [8]. The PKLot dataset comprises 12,417 images of parking lots and an impressive 695,899 images of segmented parking spaces. The dataset incorporates images captured under various weather conditions, including sunny, cloudy, and rainy days, ensuring the model’s robustness to weather variations. Images were collected at different times of the day, including diverse lighting conditions that a real-world parking detection system would encounter. The dataset was acquired from the parking lots of two Brazilian universities: the Federal University of Parana (UFPR) and the Pontificial Catholic University of Parana (PUCPR), both located in Curitiba, Brazil. Investigations have revealed that UFPR04 presents a slightly greater challenge than the other two subsets, UFPR05 and PUCPR; this is because this subset contains images with different obstacles and ground patterns. The dataset includes both occupied and empty parking spaces, allowing for comprehensive classification tasks. The dataset contains images of parking lots with delimited spaces, both occupied and empty. Figure 8 shows some examples from all three different camera points in different weather conditions.

## 4. Proposed Method

This section examines the development process of the deep learning-based parking lot occupancy detection system and its constituent components. We use the LeakyReLU6 activation function for the shallow part of the model, replace the SE block with a convolution block attention module, and replace the depth-wise convolution layers with blueprint separable convolutions. The logical architecture of the occupancy detection process with an already trained model is presented in Algorithm 1.
**Algorithm 1.** Pseudocode for parking lot occupancy detection process.
Input: images of streaming cameraInput: manually entered parking space locationsSet classification threshold → TWhen the streaming video does not stop, for each frame of the video:
Divide frame into patches according to manually predefined locationsResize the patchesFor each patch:i.Feed it to the trained modelii.Obtain its classification resultiii.If the classification result is higher than threshold T, mark it as occupied, else vacantiv.Draw bounding box around patch in the frame in red color if it is occupied, else in green colorEnd for cycle.Show the frame with bounding boxes drawn over the initial frame
End while

### 4.1. LeakyReLU6 Activation Function for the Shallow Part of the Network

The use of activation functions is an important aspect of deep learning models. Activation functions introduce non-linearity into the network, thereby allowing it to learn more complex and abstract features from the input data. The authors of MobileNetV3 used the ReLU6 activation function as part of the h-swish activation function. ReLU6 is a popular activation function that is frequently deployed in neural networks because it is computationally efficient and can prevent the vanishing gradient problem.
ReLU6(x) = min(max(0, x), 6) (3)

However, ReLU6 has the limitation that it remains inactive for negative input values, which can result in inaccurate feature extraction. To address this limitation, the Leaky-ReLU6 activation function is used in this study. The Leaky-ReLU6 function combines the leaky-ReLU concept with the ReLU6 function to form a new activation function that is divided into three segments.

When x is less than zero, the function is multiplied by a small parameter, ‘a’, to prevent the neuron from dying. This allows for more effective feature extraction in the low-level network. When 0 < x < 6, the function grows linearly; when x reaches 6, it remains at 6 and does not increase further.
Leaky-ReLU6(x) = min(6, max(ax, x)) (4)

The use of Leaky-ReLU6 in the shallow part of the MobileNetV3 model can help improve the accuracy of image feature extraction, particularly for negative input values. The parameter ‘a’ can be manually adjusted during the training process to find the optimal value for the best performance; this value can be used in subsequent test executions.

During our experiments, we tested values in the range [0.0001:0.1]. When a was equal to 0.001, the observed performance was better than the other experimental values.

### 4.2. CBAM Attention Mechanism

In computer vision, the attention mechanism is a technique that focuses on specific regions of an image that are most relevant to a given task or objective. It is inspired by the manner in which human attention works, where we tend to focus on the most informative or interesting parts of an image. In an attention mechanism, a model learns to assign importance weights to different parts of an image and then selectively combines these features to make a prediction or decision. This can improve the accuracy and efficiency of a model because it allows it to pay attention to the most important details while avoiding unimportant or distracting details in a picture. Attention mechanisms have been demonstrated to enhance the performance of these models in several computer-vision tasks, including image classification, object identification, and image captioning.

The attention module in MobileNetV3 is called the squeeze-and-excitation (SE) module. It comprises two main operations: squeeze and excitation. In the squeeze operation, the feature maps from the previous convolutional layer are globally averaged and pooled to produce a 1D feature vector that represents the channel-wise statistics of the feature maps. During the excitation operation, this 1D feature vector is passed through two fully connected layers using a gating mechanism, producing a channel-wise importance score vector. This vector is then multiplied with the original feature maps to produce the attended feature maps, which emphasize the informative channels and suppress the less informative ones.

The SE module is designed to adaptively adjust the channel-wise importance of feature maps, which enhances the discriminability of features and boosts the performance of object-detection tasks. It has been shown to perform well in a range of computer vision tasks such as semantic segmentation, object detection, and image classification. However, the SE module concentrates solely on the channel dimension of the feature map while overlooking the spatial dimension of the target data. In contrast, the convolution block attention module (CBAM) creates an attention map in both the channel and spatial dimensions and conducts element-wise multiplication operations between the attention map and input feature map in the corresponding dimensions. This results in a more comprehensive and accurate extraction of the target features.

The CBAM channel attention mechanism is characterized by a greater number of parallel global max pooling layers than the SE module. In addition, the utilization of diverse pooling operations enables the extraction of more comprehensive, high-level features. Within the bottleneck structure of the parking space classification model, the input channels undergo a dimensional upgrade and deep convolution, obtaining feature F through deep convolution; this feature is input into the channel attention module of the CBAM to derive the channel feature. The resulting channel feature F’ is then multiplied with F to obtain the feature F’, which is fed into the spatial attention module to produce the spatial feature. The final feature F’’ is obtained by multiplying the channel feature F’ and the spatial feature, followed by linear point-by-point convolution. Figure 9 shows a schematic diagram of MobileNetV3’s bottleneck structure with an integrated CBAM module.

### 4.3. Blueprint Separable Convolutions to Replace Depth-Wise Separable Convolutions

As discussed in Section 3.2, depth-wise separable convolutions are used in MobileNetV3 to reduce the number of parameters and computational complexity while maintaining accuracy. Traditional convolutional layers have a large number of parameters, which can lead to slow inference times and high memory usage. In MobileNetV3, the use of depth-wise separable convolutions, along with other optimizations, such as SE blocks and hard-swish activation functions, results in a highly efficient and accurate neural network architecture for mobile and embedded devices. However, Haase and Amthor [6] quantitatively analyzed the properties of kernel weights obtained from trained models and found that depth-wise separable convolutions indirectly rely on correlations between kernels; however, their proposed new approach, blueprint separable convolutions, utilizes intra-kernel correlations to enable a more effective separation of standard convolutions, as opposed to traditional convolutional neural networks that rely on inter-kernel correlations. This results in a more efficient and effective convolution method.

Blueprint separable convolutions are a type of convolutional neural network layer introduced by Haase and Amthor [6] that aims to improve the efficiency of depth-wise separable convolutions by exploiting the interrelationships between CNN kernels along their depth dimension. Depth-wise separable convolutions employ M × K × K filters that can be represented by a K × K template and M parameters that distribute the template in the depth dimension; this observation has motivated the creation of blueprint-separable convolutions. Every filter kernel F(n) can be depicted using a blueprint B(n) and the weights wn, 1, …, wn, M via
F(n)m,:,: = wn, m * B(n) (5)
with m in {1, …, M: number of kernels in one filter} and n in {1, …, N: number of filters in one layer}. Figure 10 illustrates the blueprint separable convolutions and their differences from standard convolutions. Blueprint separable convolutions exploit the CNN kernel correlations along their depth axes. Consequently, each filter kernel is represented as a single two-dimensional blueprint kernel in blueprint separable convolutions, which are then distributed along the depth axis using a weight vector. Although filter kernels are subject to strict limitations under this formulation, the authors experimentally showed that, when compared to their vanilla equivalents, CNNs trained using blueprint separable convolutions can achieve the same or even higher quality.

Compared to standard convolution layers that have M×N×K2 free parameters, blueprint separable convolution only has N×K2 parameters for the blueprints and M × N parameters for the weights. The authors proposed two versions of blueprint separable convolutions: unconstrained blueprint separable convolutions (BSConv-U) and subspace blueprint separable convolutions (BSConv-S).

When compared to DSConv, BSConv-U has depth-wise and point-wise convolution layers in opposite order, in which intra-kernel correlations are promoted more than cross-kernel correlations. BSConv-U is less complex in terms of the mathematical equations and calculations, making it more suitable for practical implementation.

Reversing the order of the layers is not expected to significantly affect the middle flow of the network because it already includes point-wise and depth-wise convolutions in an alternating pattern. However, the entry flow is affected because the feature maps from the initial regular convolution can be more fully utilized by the depth-wise convolution via the preceding point-wise distribution. The authors experimentally demonstrated that CNNs trained using the BSConv method can achieve comparable or even superior quality compared to their conventional counterparts.

Overall, the improvements in the architecture of the proposed model helped prevent the model from overfitting, decreased the inference time, and improved accuracy.

### 4.4. Implementation Details

The proposed classification model was trained using a personal computer with an 8-core 3.70 GHz CPU, 32 GB Memory, and Nvidia GeForce RTX 3060 GPU. The training and testing processes utilized two commonly used parking lot datasets: PKLot and CNRPark-EXT. During our experiments, we used predefined training, validation, and testing subsets of the CNRPark-EXT dataset: the training subset contains 104,493 patches from both the CNRPark and CNRPark-EXT dataset training subsets; the validation subset contains 21,231 patches from both the CNRPark and CNRPark-EXT datasets; and the testing subset contains 31,825 patches from the CNRPark-EXT dataset testing subset. From the PKLot dataset, we used the PUCPR (424,269 patches), UFPR04 (105845 patches), and UFPR05 (165,785 patches) subsets alternatively as our training and testing subsets. The crucial parameters for the training experiments are as follows: 500 epochs, a batch size of 64 images, and a 224 × 224 input image size. Using a starting learning rate of 0.0001, weight decay of 0.0005, and momentum of 0.99, we employed the Adam optimizer, which combines the benefits of two other optimizers: the adaptive gradient algorithm (AdaGrad) and root mean square propagation (RMSProp).

Using five-fold cross-validation, we separated the dataset into five sections and used 80% of it for training and the remaining 20% for validation throughout the training phase. Shuffling was performed at every epoch. Our trained model performed well when tested on an untested sample of photographs.

We used accuracy and AUC scores as our main metrics in this work. Below, we present the formulas used to calculate the accuracy and precision:Accuracy: used to evaluate the performance of the identification task. It is calculated as the number of all correct predictions divided by the total number of the dataset and the best accuracy is 1.0, which is calculated as follows:
Accuracy = (TP + TN)/(TP + TN + FN + FP)(6)
where TP, FN, FP, and TN represent the number of true positives, false negatives, false positives, and true negatives, respectively.

AUC score: a metric commonly used to evaluate the performance of binary classification models, such as those used in machine learning and deep learning. The receiver operating characteristic (ROC) curve is a graphical representation that illustrates the trade-off between the true positive rate (sensitivity) and the false positive rate (1 specificity) at different probability thresholds. The AUC represents the area under the ROC curve, which is a single value ranging from 0 to 1. The AUC score in our work measures the model’s ability to distinguish between occupied and unoccupied parking spaces.

## 5. Experimental Results and Analysis

In this section, we analyze and compare the results of our proposed model with those of other classification models developed for parking lot classification, such as mAlexNet, CarNet, and others, in terms of classification accuracy and AUC score. The experiments show that our proposed modified MobileNetV3 model has a higher classification accuracy than other models and that our proposed model correctly classifies and categorizes more empty and busy parking spaces than other models.

We tried to visualize what our model learnt during the training process and used GradCAM [22] and feature visualization [23] methods to check if our model was learning the right features and paying attention to the right part of the image. In Figure 11, samples are given for this process. GradCAM helps by visualizing which parts of the image the model is paying the most attention to.

In Figure 12, we demonstrate the sample parking lot classification result performed with our proposed model. As is visible in the figure, all the parking spaces are correctly classified as busy or vacant, which shows the accuracy of our model.

As an ablation study, we trained the original MobileNetV3 model from scratch on PKLot and CNRPark-EXT datasets and tested the model on both datasets, and the same process was applied to four different models: MobileNetV3 with the proposed LeakyReLU6 activation function, MobileNetV3 with its SE mechanism replaced by the CBAM attention mechanism, MobileNetV3 with its depth-wise separable convolutions replaced by blueprint separable convolutions, and MobileNetV3 with all the above modifications applied. The goal of these experiments was to detect which modification made to the original model brought the greatest increase in accuracy and made the model more generalized and scalable to different parking areas. The results are summarized in Table 3.

From Table 3, it is evident that although the original MobileNetV3 model achieved nearly 100% accuracy on the same training and testing subsets of the PKLot dataset. But, when trained on one subset and tested on another, the accuracy of this model dropped, which means that it overfit the dataset. When the model was trained on the UFPR05 dataset and tested on two different subsets, its performance was not good, achieving accuracy rates of 87.80% for PUCPR testing and 88.25% for UFPR05 testing. However, changing its shallow part activation function, changing its attention mechanism, and replacing depth-wise separable convolutions with blueprint separable convolutions helped the model avoid overfitting and achieve high accuracy on all training and testing parts.

Substituting the ReLU6 activation function with LeakyReLU6 resulted in a reduction in overfitting of approximately 2% within identical training and testing dataset scenarios. Introducing the CBAM module in lieu of the SE module led to a noteworthy accuracy enhancement from 87.80% to 92.64% for the UFPR05/PUCPR case and from 88.25% to 91.78% for the UFPR05/UFPR04 scenario. Conversely, replacing DSConv with BSConv yielded the most significant improvement in accuracy among the three architectural modifications. In the case of training and testing on the same subset, the accuracy nearly approximated that of the original MobileNetV3, while successfully mitigating overfitting. Moreover, for the UFPR05/PUCPR and UFPR05/UFPR04 cases, the model’s accuracy exhibited improvements of 6% and 5%, respectively. The best classification results were achieved when all modifications were applied to the model, which was expected regarding the modifications to the model structure and their effects on the model’s performance.

Figure 13 presents the learning curves of five different models in Table 3 for training on the PUCPR subset of the PKLot dataset.

In Figure 13, it can be seen that after the final epoch, the training accuracies for the original MobileNetV3 and our proposed approach (MobileNetV3 with all modifications) were 99.95% and 99.9%. Also, this comparison shows that out of all three architectural changes, replacing DSConv with BSConv had more effect on the model’s classification improvement. However, as it was said before, the original MobileNetV3 overfitted the dataset, so it achieved higher accuracy compared to the one we proposed.

We then compared the results of our best model with those of other models developed or fine-tuned with transfer learning, such as AlexNet, mAlexNet, CarNet, VGG16 [24], VGG19 [24], and others, on the PKLot dataset. A comparison of the results is presented in Table 4.

**Table 4 sensors-23-07642-t004:** Classification results comparison of our best model with mAlexNet, CarNet, VGG16, and other models on PUCPR, UFPR04, UFPR05 subsets of PKLot [8] dataset. Bold data shows the highest score for that experiment.

Model	Train	Test		
		PUCPR	UFPR04	UFPR05
Our solution: modified MobileNetV3	PUCPR	**99.90%**	**98.20%**	95.15%
	UFPR04	**98.85%**	**99.68%**	**98.38%**
	UFPR05	95.06%	**96.34%**	99.20%
CarNet [16]	PUCPR	98.80%	94.40%	**97.70%**
	UFPR04	98.30%	95.60%	97.60%
	UFPR05	**98.40%**	95.20%	97.50%
mAlexNet [7]	PUCPR	**99.90%**	98.03%	96%
	UFPR04	98.27%	99.54%	93.29%
	UFPR05	92.72%	93.69%	**99.49%**
AlexNet [14]	PUCPR	98.60%	88.80%	83.40%
	UFPR04	89.50%	98.20%	87.60%
	UFPR05	88.20%	87.30%	98%
VGG16 [24]	PUCPR	88.20%	94.20%	90.80%
	UFPR04	89.70%	95.30%	90%
	UFPR05	90.50%	94.90%	91.80%
VGG19 [24]	PUCPR	81.50%	93.80%	94.60%
	UFPR04	80.40%	92.30%	91.90%
	UFPR05	88.80%	95.10%	95.90%
Xception [25]	PUCPR	96.30%	92.50%	93.30%
	UFPR04	94%	94.60%	93.40%
	UFPR05	95.70%	90.90%	91.20%
Inception V3 [26]	PUCPR	90.80%	91.10%	94.20%
	UFPR04	91.70%	95.20%	92.40%
	UFPR05	94.30%	92.90%	93.70%
ResNet50 [27]	PUCPR	88.20%	94.20%	94.10%
	UFPR04	89.70%	95.30%	93.30%
	UFPR05	90.50%	94.90%	95.50%

The results presented in Table 4 indicate that our approach demonstrated superior performance compared to the alternative classification methods across six out of nine experimental scenarios. Notably, our method exhibited higher accuracy rates in the following scenarios: PUCPR/PUCPR (99.9%), PUCPR/UFPR04 (98.2%), UFPR04/PUCPR (98.85%), UFPR04/UFPR04 (99.68%), UFPR04/UFPR05 (98.38%), and UFPR05/UFPR04 (96.34%). Notably, CarNet [16] exhibited better performance than our proposed model in the UFPR05/PUCPR and PUCPR/UFPR05 scenarios, recording accuracy rates of 98.4% compared to 95.06% and 97.7% compared to 95.15%, respectively. Additionally, in the UFPR05/UFPR05 scenario, mAlexNet [7] achieved the highest accuracy of 99.49%, whereas our model attained an accuracy of 99.2%. These results show that the modifications to the original MobileNetV3 model are as useful and efficient as expected.

We subsequently repeated the experiments using the CNRPark-EXT dataset. First, we trained five models on the training subset of the CNRPark-EXT and tested them on the testing subset of the dataset: original MobileNetV3, MobileNetV3 with the LeakyReLU6 activation function, MobileNetV3 with the CBAM module, MobileNetV3 with BSConv, and MobileNetV3 with all architecture modifications. The results of these experiments are presented in Table 5.

The initial MobileNetV3 architecture yielded accuracies of 94.95%, 90.13%, and 93.53% on the training, validation, and testing subsets of the dataset, respectively. The introduction of an alternative activation function resulted in a modest enhancement of approximately 0.5% in accuracy. Meanwhile, the adoption of an alternative attention module led to a notable improvement of 2% in accuracy. Substitution of depth-wise separable convolutions (DSConv) with blueprint separable convolutions (BSConv) yielded a substantial increase of about 2.5% in accuracy.

Figure 14 shows the training process for the five different models in Table 5 on the training subset of the CNRPark-EXT dataset.

From Figure 14, it is visible that, as expected, the architectural changes helped the model increase its accuracy. In this dataset, the changes with the biggest accuracy increase were replacing the SE module with the CBAM module and replacing DSConv with BSConv.

After finishing the experiment with different modifications, we compared our best model results with those of the CarNet, AlexNet, and ResNet models on the CNRPark-EXT dataset. A comparison of the results is presented in Table 6. From Table 6, we can observe that our model performed better in two out of three tasks in the training and testing subsets of the CNRPark-EXT dataset. Our model’s validation result was also good but slightly lower than that of AlexNet. Our model achieved 97.73% accuracy for the validation subset; AlexNet achieved 97.91% accuracy. The previous state-of-the-art model, CarNet, achieved 97.91% accuracy in the training subset of the dataset, while achieving 90.05% and 97.24% accuracies in the validation and test sets of the dataset.

Finally, we compared our best model with mAlexNet and AlexNet in combination with the CNRPark EXT and PKLot datasets. The test results are provided in Table 7.

As CarNet was specifically designed for this task, it achieved 97.03% accuracy on average for all three different experiments. AlexNet obtained 94.07% accuracy as it is a good general deep learning architecture. However, mAlexNet achieved only 88.69% accuracy on average for all three different experiments, which shows that mAlexNet achieves very poor results when trained on one full dataset and tested on another, or in the reverse case. The testing scores for the three combinations provided reveal that our model is much more robust, as it can generalize well and learn general features from the datasets.

In Table 8, the AUC scores for our proposed model and other state-of-the-art models are given and compared. In this table, we include one different model proposed in [8], which we call PKLot for convenience. Out of nine experiments with different subsets of the PKLot dataset, our proposed model achieved the highest AUC scores in five cases, while the PKLot approach had the highest AUC scores in three experiments, and CarNet achieved the highest AUC score in one experiment when trained on the PUCPR subset and tested on the UFPR05 subset of the PKLot dataset.

Our trained models took around 10 MB memory, which is quite good compared to big models like VGG16, AlexNet, etc. A modified version of mAlexNet proposed in [15] needs about 10 KB memory, but its accuracy is lower than mAlexNet. mAlexNet, proposed by Amato et al. [7], needed about 129 KB. So, while our model is bigger than mAlexNet and modified mAlexNet in size, it has better accuracy and AUC score, as shown in the above experiments.

We also compared the average runtimes of our proposed model with those of other models. We randomly selected 1000 224 × 224 images from each of the CNRPark-EXT and PKLot datasets and ran each model on the same machine used for training without GPU acceleration in the PyTorch framework. Table 9 shows our runtime analysis.

While our model is 6.7 times slower than both mAlexNet and custom mAlexNet models, it is still 3 times faster than the AlexNet model, which makes it applicable in real-world applications.

The overall conclusion is that the improved MobileNetV3 is a fairly robust model when trained on one dataset and tested on another. We are certain that this approach can be applied to real-life scenarios.

## 6. Conclusions and Future Work

A parking lot occupancy detection approach was developed in this study using a deep CNN classification model, MobileNetV3, with several modifications to its architecture that increased its robustness and accuracy. The developed model was trained on two well-known parking lot datasets: PKLot and CNRPark-EXT. The incoming video stream is processed frame-by-frame, and each frame is split into patches; the modified MobileNetV3 model classifies each patch as being occupied by a car or as an empty parking space. The classification results were integrated into frames with bounding boxes drawn around each parking space. The qualitative and quantitative performances of the proposed system were experimentally compared with those of other established classification models. The evaluation and experimental results revealed that the enhanced MobileNetV3 model achieved high accuracy and outperformed the other classification models in terms of both accuracy and speed. The developed parking-space classification model is efficient and can be applied to real-world scenarios using mobile devices, resource-constrained edge devices, and cameras.

The main contributions of this work are provided below:An optimal deep learning model was developed to classify parking lot spaces as empty or busy. In the proposed model, the activation function in the shallow part of the model, which requires significant calculations, is replaced by a new activation function that requires less computation. The squeeze-and-excitation attention mechanism applied in the original MobileNetV3 is replaced by another, more effective attention mechanism: the convolutional block attention mechanism. Moreover, because of the hidden cross-kernel correlations in depth-wise separable convolutions, blueprint separable convolutions are used, as they require less computation because they have fewer parameters.Using the improved MobileNetV3 model, the parking lot occupancy detection approach can precisely detect the number of free and busy parking spaces, despite different weather conditions, lighting, and shadows.

Despite being robust and sufficiently quick for real-world applications, our model still has some shortcomings: the inability to correctly classify images under diverse weather conditions, images that contain a portion of cars, images with unusual parking configurations, images with partial occlusion, and images with unseen objects.

In the future, we plan to continue exploring new methods and changes to improve the accuracy of the classification model, reduce its runtime to make it faster when applied to mobile and edge devices, and make it successfully applicable in the above mentioned cases where the model may fail now. Furthermore, we plan to work on a smart camera containing the proposed system to detect parking lot occupancy, improve its efficiency, and reduce its resource consumption.

Our research can be extended by being integrated into a decentralized smart camera system [7]. Incorporating the Improved MobileNetV3 into a decentralized smart camera system has the potential to significantly enhance the efficiency, responsiveness, and intelligence of the system. Also, we are working on automatic parking space detection to replace manually labeling the parking spaces.

## Figures and Tables

**Figure 1 sensors-23-07642-f001:**
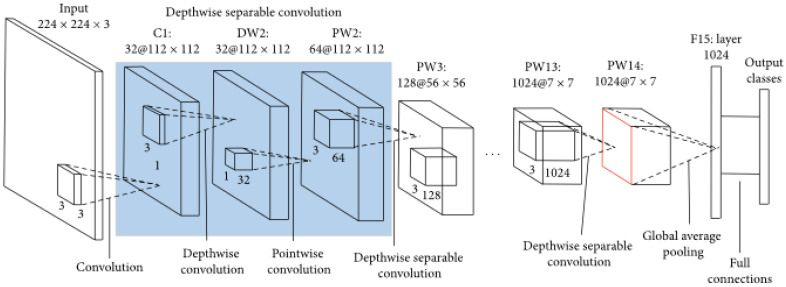
Architecture of MobileNetV1. C1—convolution layer 1, DW2—depth-wise convolution 2, PW2—point-wise convolution 2, F15—fully connected layer 15.

**Figure 2 sensors-23-07642-f002:**
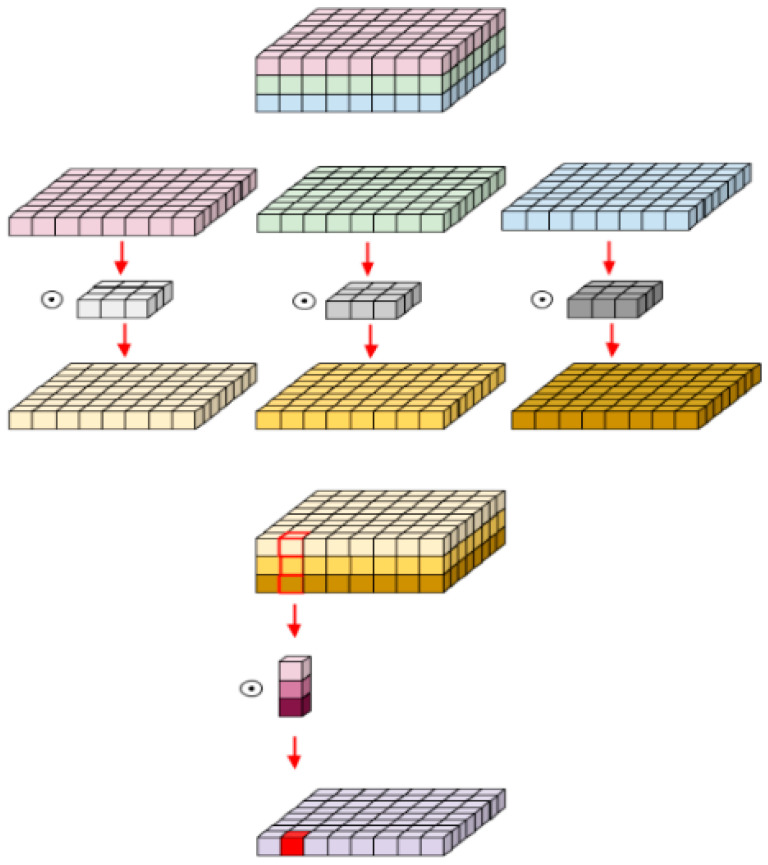
Depth-wise separable convolution. First, filters are applied per channel, and then the outputs are convolved with 1 × 1 filter to reduce the depth dimension.

**Figure 3 sensors-23-07642-f003:**
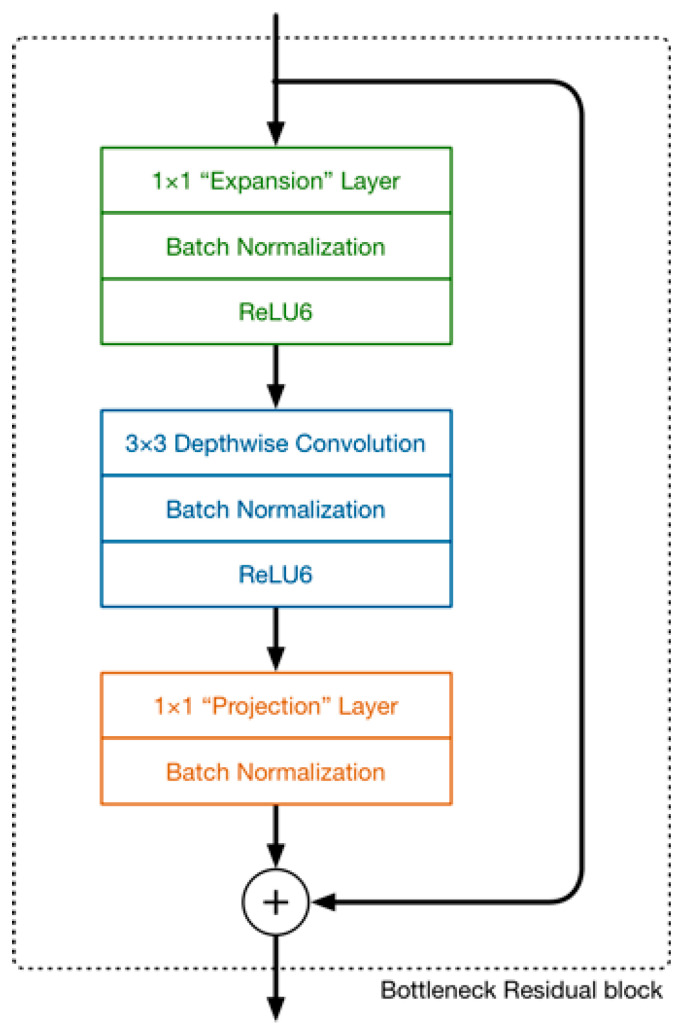
Main building block of MobileNetV2.

**Figure 4 sensors-23-07642-f004:**
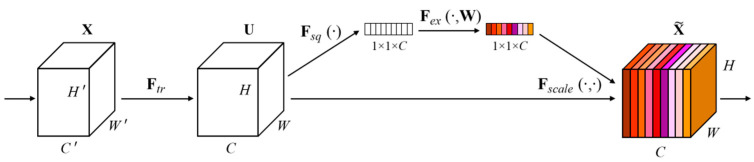
Architecture of squeeze-and-excitation block proposed within SENet model.

**Figure 5 sensors-23-07642-f005:**
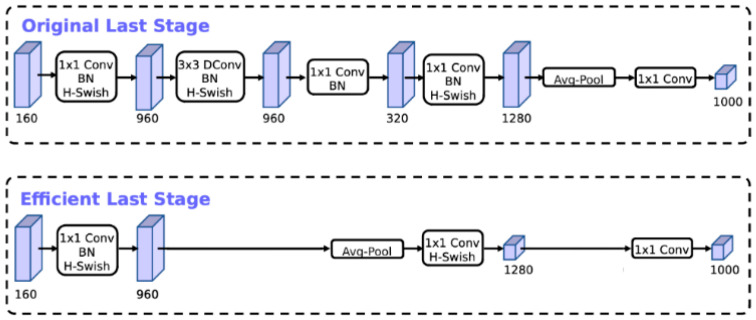
Comparison of the original and efficient last stages. The efficient last stage can drop three expensive layers with no loss of accuracy. BN—bottleneck layer, Dconv—depth-wise separable convolution, Avg-pool—average pooling.

**Figure 6 sensors-23-07642-f006:**
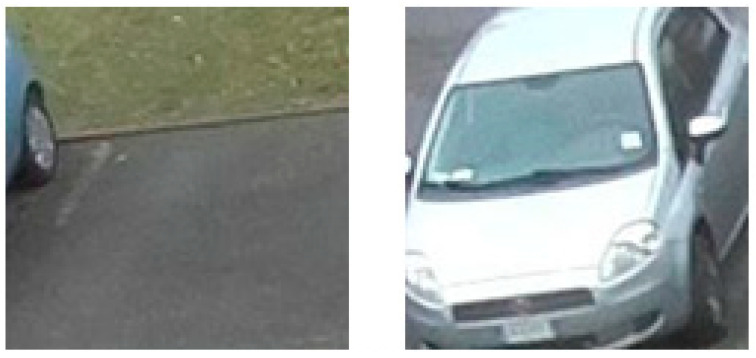
Empty and busy parking spaces.

**Figure 7 sensors-23-07642-f007:**
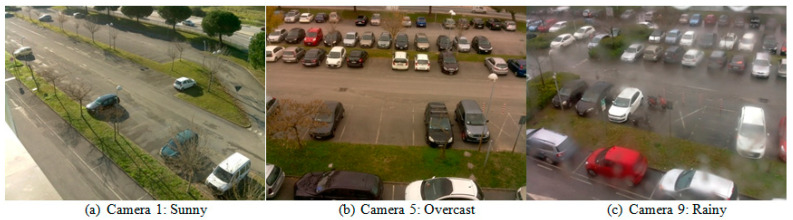
CNRPark-EXT dataset samples: (**a**–**c**) taken in 3 weather conditions.

**Figure 8 sensors-23-07642-f008:**
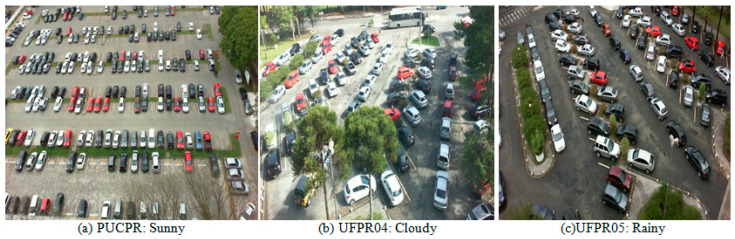
PKLot dataset samples. (**a**–**c**) show examples of different parking lots and weather
conditions.

**Figure 9 sensors-23-07642-f009:**
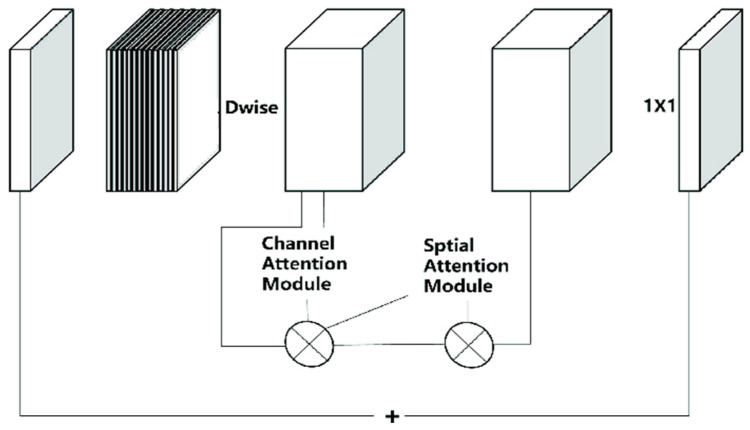
The structure diagram of MobileNetV3′s bottleneck layer structure after adding CBAM. Dwise—depth-wise convolution.

**Figure 10 sensors-23-07642-f010:**
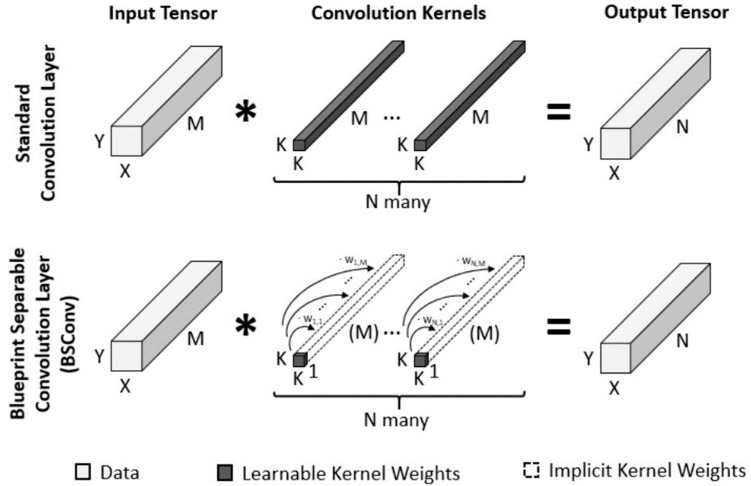
Blueprint separable convolution.

**Figure 11 sensors-23-07642-f011:**
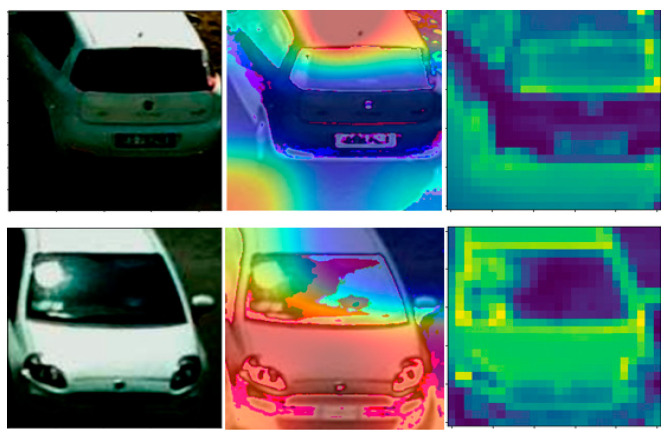
Images on the left are busy parking space patches; images in the middle are taken with GradCAM; images on the right are extracted with the first layer of our trained model.

**Figure 12 sensors-23-07642-f012:**
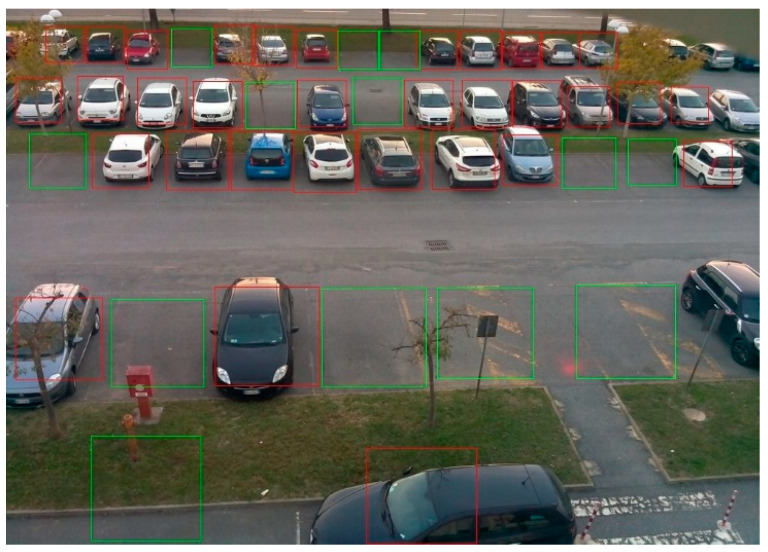
Sample parking lot visualization result with our proposed model.

**Figure 13 sensors-23-07642-f013:**
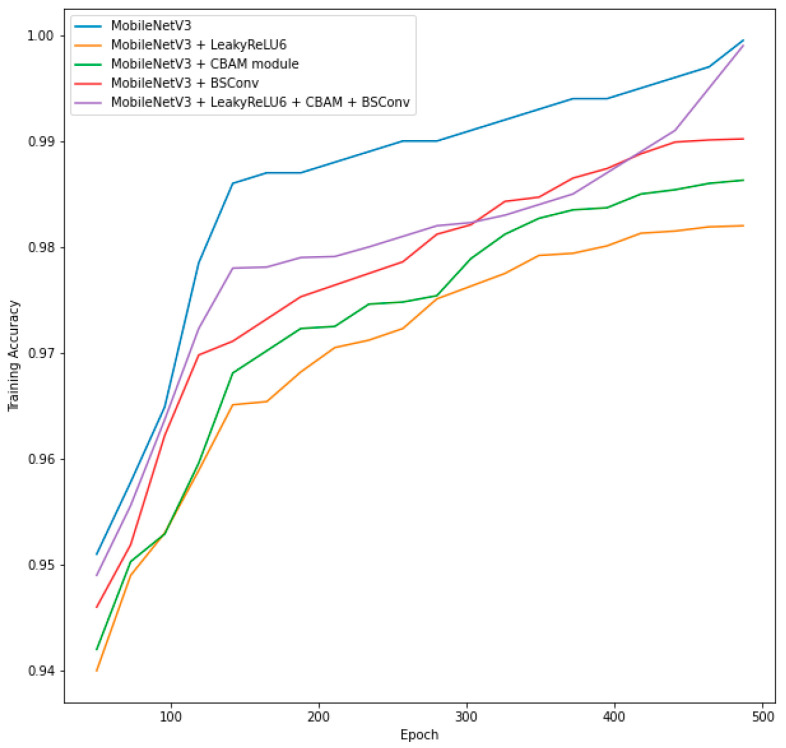
Comparison of 5 different model training processes on PUCPR subset of PKLot dataset. x-axis: number of epochs (500), y-axis: accuracy.

**Figure 14 sensors-23-07642-f014:**
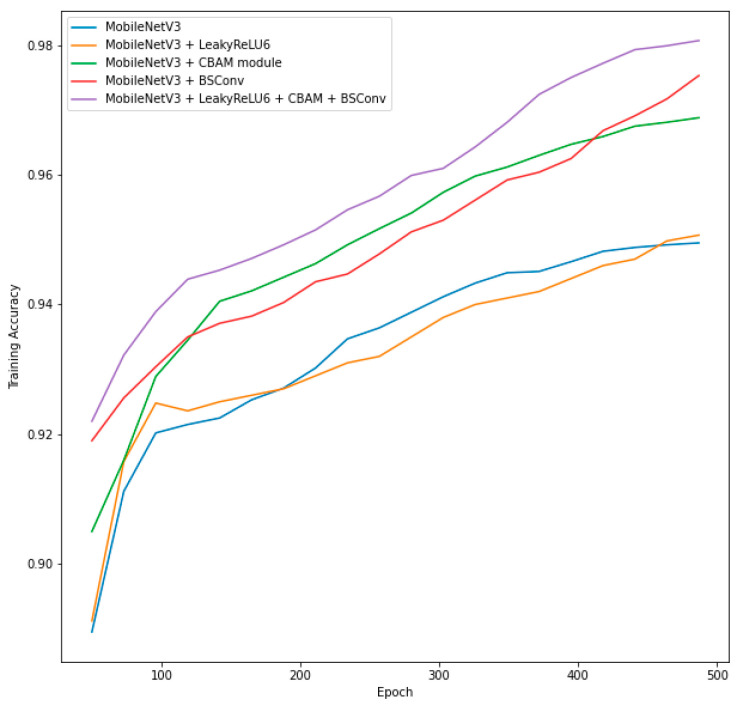
Comparison of 5 different model training processes on training subset of CNRPark-EXT dataset. x-axis: number of epochs (500), y-axis: accuracy.

**Table 1 sensors-23-07642-t001:** Specification for MobileNetV3-Large and MobileNetV3-Small. SE means whether there is a squeeze-and-excitation in this block; NL is the type of non-linearity used; HS means h-swish; RE is ReLU; NBN means no batch normalization; S is stride; in SE columns + means SE is used in this layer, - means it is not used.

MobileNetV3-Large							MobileNetV3-Small						
Input	Operator	Exp Size	Out channels	SE	NL	s	Input	Operator	Exp Size	Out channels	SE	NL	s
224^2^ × 3	conv2d, 3 × 3	-	16	-	HS	2	224^2^ × 3	conv2d, 3 × 3	-	16	-	HS	2
112^2^ × 16	bneck, 3 × 3	16	16	-	RE	1	112^2^ × 16	bneck, 3 × 3	16	16	+	RE	2
112^2^ × 16	bneck, 3 × 3	64	24	-	RE	2	56^2^ × 16	bneck, 3 × 3	72	24	-	RE	2
56^2^ × 24	bneck, 3 × 3	72	24	-	RE	1	28^2^ × 24	bneck, 3 × 3	88	24	-	RE	1
56^2^ × 24	bneck, 5 × 5	72	40	+	RE	2	28^2^ × 24	bneck, 5 × 5	96	40	+	HS	2
28^2^ × 40	bneck, 5 × 5	120	40	+	RE	1	14^2^ × 40	bneck, 5 × 5	240	40	+	HS	1
28^2^ × 40	bneck, 5 × 5	120	40	+	RE	1	14^2^ × 40	bneck, 5 × 5	240	40	+	HS	1
28^2^ × 40	bneck, 3 × 3	240	80	-	HS	2	14^2^ × 40	bneck, 5 × 5	120	48	+	HS	1
14^2^ × 80	bneck, 3 × 3	200	80	-	HS	1	14^2^ × 48	bneck, 5 × 5	144	48	+	HS	1
14^2^ × 80	bneck, 3 × 3	184	80	-	HS	1	14^2^ × 48	bneck, 5 × 5	288	96	+	HS	2
14^2^ × 80	bneck, 3 × 3	184	80	-	HS	1	7^2^ × 96	bneck, 5 × 5	576	96	+	HS	1
14^2^ × 80	bneck, 3 × 3	480	112	+	HS	1	7^2^ × 96	bneck, 5 × 5	576	96	+	HS	1
14^2^ × 112	bneck, 3 × 3	672	112	+	HS	1	7^2^ × 96	conv2d, 1 × 1	-	576	+	HS	1
14^2^ × 112	bneck, 5 × 5	672	160	+	HS	2	7^2^ × 576	pool, 7 × 7	-	-	-	-	1
7^2^ × 160	bneck, 5 × 5	960	160	+	HS	1	1^2^ × 576	conv2d 1 × 1, NBN	-	1024	-	HS	1
7^2^ × 160	bneck, 5 × 5	960	160	+	HS	1	1^2^ × 1024	conv2d 1 × 1, NBN	-	k	-	-	1
7^2^ × 160	conv2d, 1 × 1	-	960	-	HS	1							
7^2^ × 160	pool, 7 × 7	-	-	-	-	1							
1^2^ × 960	conv2d 1 × 1, NBN	-	1280	-	HS	1							
7^2^ × 1280	conv2d 1 × 1, NBN	-	k	-	-	1							

**Table 2 sensors-23-07642-t002:** Main features of CNRPark-EXT [7] and PKLot [8] datasets.

Dataset	Image Resolution	SAMPLE Time	Number of Cameras/Parking Lots	Number of Images	Number of Annotations (Number of Occupied/Number of Empty)
CNRPark-EXT	1000 × 750 px	30 min	9	4278	157,549 (87,709/69,840)
PKLot	1280 × 720 px	5 min	3	12,417	695,899 (337,780/358,119)

**Table 3 sensors-23-07642-t003:** Performance results of all modified versions of MobileNetV3 model on PUCPR, UFPR04, UFPR05 subsets of PKLot [7] dataset. Bold data shows the highest score for that experiment.

Model	Train	Test		
		PUCPR	UFPR04	UFPR05
MobileNetV3	PUCPR	**99.95%**	96.47%	91.02%
	UFPR04	98.30%	**99.95%**	95.56%
	UFPR05	87.80%	88.25%	**99.88%**
MobileNetV3 + LeakyReLU6	PUCPR	98.20%	94.68%	89.95%
	UFPR04	96.65%	97.80%	93.40%
	UFPR05	89.95%	90.45%	97.86%
MobileNetV3 + CBAM module (convolution block attention module)	PUCPR	98.65%	95.34%	91.27%
	UFPR04	97.80%	98.33%	94.28%
	UFPR05	92.64%	91.78%	98.14%
MobileNetV3 + BSConv (blueprint separable convolutions)	PUCPR	99.03%	96.13%	92.07%
	UFPR04	98.57%	99.18%	95.69%
	UFPR05	93.85%	93.63%	99.17%
MobileNetV3 + LeakyReLU6 + CBAM + BSConv	PUCPR	99.90%	**98.20%**	**95.15%**
	UFPR04	**98.85%**	99.68%	**98.38%**
	UFPR05	**95.06%**	**96.34%**	99.20%

**Table 5 sensors-23-07642-t005:** Performance results of all modified versions of MobileNetV3 on CNRPark-EXT [7] dataset. Bold data shows the highest score for that experiment.

Model	Training	Validation	Testing
MobileNetV3	94.95%	90.13%	93.53%
MobileNetV3 + LeakyReLU6	95.07%	90.28%	94.09%
MobileNetV3 + CBAM module (convolution block attention module)	96.88%	91.99%	95.03%
MobileNetV3 + BSConv (blueprint separable convolutions)	97.53%	92.89%	95.86%
MobileNetV3 + LeakyReLU6 + CBAM + BSConv	**98.07%**	**97.73%**	**97.69%**

**Table 6 sensors-23-07642-t006:** Classification results comparison of our best model with CarNet, AlexNet, and ResNet50 on CNRPark-EXT [7] dataset. Bold data shows the highest score for that experiment.

Model	Training	Validation	Testing
Our solution: modified MobileNetV3	**98.07%**	97.73%	**97.69%**
CarNet	97.91%	90.05%	97.24%
AlexNet	96.99%	**97.91%**	96.54%
ResNet50	96.51%	97.80%	96.24%

**Table 7 sensors-23-07642-t007:** Comparison of results of our model with CarNet and mAlexNet in combination of PKLot [8] and CNRPark EXT [7]. Bold data shows the highest score for that experiment.

Model	Training Dataset	Testing Dataset	Accuracy (%)	Mean (%)
Our solution: modified MobileNetV3	PKLot	CNRPark EXT	**96.38%**	
	CNRPark EXT	PKLot	**98.49%**	**98.01%**
	CNRPark EXT	CNRPark EXT	**99.17%**	
CarNet	PKLot	CNRPark EXT	94.77%	
	CNRPark EXT	PKLot	98.21%	97.03%
	CNRPark EXT	CNRPark EXT	98.11%	
mAlexNet	PKLot	CNRPark EXT	83.83%	
	CNRPark EXT	PKLot	84.53%	88.69%
	CNRPark EXT	CNRPark EXT	97.71%	
AlexNet	PKLot	CNRPark EXT	90.52%	
	CNRPark EXT	PKLot	93.70%	94.07%
	CNRPark EXT	CNRPark EXT	98%	

**Table 8 sensors-23-07642-t008:** Comparison of AUC scores of modified MobileNetV3 with CarNet, PKLot [8], and mAlexNet on 3 subsets of PKLot [8] dataset. Bold data shows the highest score for that experiment.

Name of Architecture	Training Dataset	Testing Dataset	AUC Score	Best Result Achieved Method
modified MobileNetV3	UFPR04	UFPR04	0.992	PKLot
CarNet	UFPR04	UFPR04	0.979	
PKLot	UFPR04	UFPR04	**0.999**	
mAlexNet	UFPR04	UFPR04	0.99	
modified MobileNetV3	UFPR04	UFPR05	**0.9956**	modified MobileNetV3
CarNet	UFPR04	UFPR05	0.9935	
PKLot	UFPR04	UFPR05	0.9595	
mAlexNet	UFPR04	UFPR05	0.99	
modified MobileNetV3	UFPR04	PUCPR	**0.9993**	modified MobileNetV3
CarNet	UFPR04	PUCPR	0.9982	
PKLot	UFPR04	PUCPR	0.9713	
mAlexNet	UFPR04	PUCPR	0.99	
modified MobileNetV3	UFPR05	UFPR04	**0.9985**	modified MobileNetV3
CarNet	UFPR05	UFPR04	0.9963	
PKLot	UFPR05	UFPR04	0.9533	
mAlexNet	UFPR05	UFPR04	0.98	
modified MobileNetV3	UFPR05	UFPR05	0.999	PKLot
CarNet	UFPR05	UFPR05	0.9989	
PKLot	UFPR05	UFPR05	**0.9995**	
mAlexNet	UFPR05	UFPR05	0.99	
modified MobileNetV3	UFPR05	PUCPR	**0.9819**	modified MobileNetV3
CarNet	UFPR05	PUCPR	0.9791	
PKLot	UFPR05	PUCPR	0.9761	
mAlexNet	UFPR05	PUCPR	0.98	
modified MobileNetV3	PUCPR	UFPR04	**0.9923**	modified MobileNetV3
CarNet	PUCPR	UFPR04	0.9845	
PKLot	PUCPR	UFPR04	0.9589	
mAlexNet	PUCPR	UFPR04	0.99	
modified MobileNetV3	PUCPR	UFPR05	0.9906	CarNet
CarNet	PUCPR	UFPR05	**0.9938**	
PKLot	PUCPR	UFPR05	0.9152	
mAlexNet	PUCPR	UFPR05	0.99	
modified MobileNetV3	PUCPR	PUCPR	0.99	PKLot
CarNet	PUCPR	PUCPR	0.9986	
PKLot	PUCPR	PUCPR	**0.9999**	
mAlexNet	PUCPR	PUCPR	0.99	

**Table 9 sensors-23-07642-t009:** Average runtime of our model with AlexNet, mAlexNet, and custom mAlexNet [28] on subsets of PKLot [8] and CNRPark EXT [7].

Model	Average Runtime on CNRPark-EXT Test (s)	Average Runtime on PKLot Test(s)	Mean Average Runtime (s)
Our solution: modified MobileNetV3	0.0587	0.0619	0.0603
AlexNet	0.181	0.182	0.1815
mAlexNet	0.009	0.009	0.009
custom mAlexNet [28]	0.009	0.009	0.009

## Data Availability

The data used in this study is openly available and can be accessed from the following sources: [7,8].

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
