# Peer review of "Parking Lot Occupancy Detection with Improved MobileNetV3"

_sensors, 2023, doi:10.3390/s23177642_

Round 1
Reviewer 1 Report
The paper delves into the realm of deep learning-based computer vision, particularly focusing on parking lot management and vehicle detection. Here are my detailed comments and suggestions:
General Comments:
Originality: The paper provides a comprehensive review and application of existing deep learning techniques in the context of parking management. While the creativity in terms of model improvement appears to be limited, the authors have done a commendable job in designing experiments and analyzing results.
Structure and Clarity: The paper is well-structured, with clear sections dedicated to literature review, methodology, experiments, and results. The flow of content is logical, making it easy for readers to follow.
Detailed Comments and Suggestions:
Introduction:
- The introduction could benefit from a clearer statement of the problem. While the application of deep learning in parking management is discussed, the specific challenges in this domain should be highlighted more prominently.
- It would be beneficial to provide a brief overview of the main contributions of the paper in the introduction.
- While the paper's title suggests an improvement on the MobileNetV3 architecture, it would be beneficial to explicitly state the limitations of the original MobileNetV3 in the context of parking lot occupancy detection. This would set a clear foundation for the proposed improvements.
Literature Review:
- The authors have provided a comprehensive list of references. However, a comparative analysis highlighting the advantages and limitations of each referenced work would add depth to the review.
- Consider reviewing the following literature as supplements:
1. Grbić, R., & Koch, B. (2023). Automatic vision-based parking slot detection and occupancy classification. Expert Systems with Applications, 225, 120147.
Grbić & Koch (2023) introduced an algorithm for automatic parking slot detection and occupancy classification. Their approach eliminates the need for manual labeling, which is a significant advancement. How does your method address this challenge? If manual labeling is still required in your approach, consider discussing potential improvements or justifications.
2. Duong, T. L., Le, V. D., Bui, T. C., & To, H. T. (2022, October). Towards an Error-free Deep Occupancy Detector for Smart Camera Parking System. In European Conference on Computer Vision (pp. 163-178). Cham: Springer Nature Switzerland.
The work by Duong et al. (2022) introduced OcpDet, an object detector for parking occupancy detection. Given that they emphasize scalability and reliability, it would be pertinent to discuss how the Improved MobileNetV3 scales in larger parking areas and its reliability in diverse conditions.
3. Martynova, A., Kuznetsov, M., Porvatov, V., Tishin, V., Kuznetsov, A., Semenova, N., & Kuznetsova, K. (2023). Revising deep learning methods in parking lot occupancy detection. arXiv preprint arXiv:2306.04288.
Martynova et al. (2023) emphasized the lack of generalization ability in existing systems. It would be beneficial to discuss how the "Improved MobileNetV3" enhances the generalization capabilities, especially when compared to the EfficientNet architecture they proposed.
Methodology:
- While the applied techniques are well-detailed, the paper could benefit from a subsection discussing the novelty or modifications made to existing models, if any.
- Elaborate on the specific improvements made to the MobileNetV3 architecture. Are these improvements architectural, related to training techniques, or both?
- Given the emphasis on "improvement," it would be beneficial to have a side-by-side comparison (in terms of performance metrics) between the original MobileNetV3 and the improved version.
Experiments and Results:
- While the paper's experimental design is solid, it would be beneficial to include ablation studies to understand the contribution of each improvement made to the MobileNetV3 architecture. In addition, more visual representations, such as graphs or charts, to depict performance metrics, are always welcome.
- Consider discussing potential failure cases or scenarios where the Improved MobileNetV3 might not perform optimally. This adds credibility and transparency to the research.
- Given the references, it would be insightful to have comparative performance metrics with architectures like YOLOv8, ResNet101, and EfficientNet on the same datasets.
Conclusion:
- Reiterate the main contributions and improvements over existing methods.
- Discuss potential extensions of the work. For instance, could the Improved MobileNetV3 be integrated into a decentralized smart camera system as proposed by Amato et al.?
- Given the advancements in vision transformers as mentioned by Martynova et al., consider discussing potential explorations in that direction.
Miscellaneous:
- Ensure that all figures and tables are clearly labeled and referenced in the text. Please check the resolution of all the figures; some of them look blurry.
- Consider providing a section or appendix with details about the datasets used, making it easier for readers or researchers to replicate the experiments.
Recommendation:
Given the solid experimental design and results analysis, I believe the paper holds value for the academic community. However, addressing the above suggestions would significantly enhance the quality and clarity of the manuscript. I recommend accepting the paper for publication after the authors make the necessary revisions based on the provided comments and suggestions.
Author Response
My response to reviewer 1 is in the attached file

Reviewer 2 Report
Review Comments
The presented work proposed an improved and optimized MobileNetV3 model trained on the CNRPark-EXT and PKLOT datasets; performance evaluation is also completed. The proposed model processes single parking space patches within the entire parking area obtained from the real-time video feed, obtains the classification results for each patch, and indicates whether they are occupied or available. However, the following corrections can be considered by the authors to further improve the quality of the manuscript.
I have some major corrections and suggestions below:-
1. The abstract need to be improved and the outcome of the work in terms of achieved various other performance calculations must be included in the abstract.
2. Paper lacks novelty as per utilized existing model. Authors must show explain the novel contribution of the work with proper justification of the outcomes. What novelty is established in this work compared to existing works? Novel contribution of the work can be added at end of introductions with proper justification of the outcomes.
3. Explaining the problem and the gaps in existing literature in a concise but self-contained way (although readers might wish to consult references, they should not be forced to do so)
4. Future work and limitations of the proposed work can be added and discussed.
5. Providing a clear yet mathematically rigorous description of the tools used.
6. Comparative analysis of various performance parameters with respect to sate of art methods must be discussed. More recent state-of-the-art approaches should be compared; the experiments should use more sizable real-world data sets from public repositories (if any);
7. Specification of the implementation platform is missing.
8. The computational complexity of the proposed work must be discussed. Also, compare the proposed method in terms of computational complexity?
9. Comparative analysis with respect to real-time time analysis is missing?
10. Literature survey is very short and narrow and need to be updated based on current state of art methods. Some more paper based on current study in Parking Lot Occupancy Detection.
11. In all results tables’/figures utilized datasets like in table 2, 3,5, 6, 7 etc. must be cited with proper and specific citations.
12. Add industrial significance of the proposed approach.
13. To make the proposed algorithm of this article more readable use pseudo-code.
14. Describing in detail the data set used and what are the expected outcomes- widening the experimental comparison including other data and methods. Utilized data sets must cited by proper sources.
15. Comparative analysis with respect to various performance metrics is missing? Comparative analysis of various performance parameters with respect to various other data sets must be discussed. The comparison can be a bit unfair if different data is not used for comparative analysis.
16. How much data should be considered for training and testing for architecture implementation? Details of training and testing data sets must be tabulated.
17. Precision vs. recall curves of the proposed algorithms with respect to data sets must be included.
Author Response
My response to reviewer 2 is in the attached word file

Round 2
Reviewer 1 Report
The author adeptly incorporated my comments. This manuscript warrants consideration for publication.
Reviewer 2 Report
All my concerns and comments has been added sucessfully. I accept it in current form.